# Effects of Phenolic-Rich *Pinus densiflora* Extract on Learning, Memory, and Hippocampal Long-Term Potentiation in Scopolamine-Induced Amnesic Rats

**DOI:** 10.3390/antiox11122497

**Published:** 2022-12-19

**Authors:** Kwan Joong Kim, Eun-Sang Hwang, Min-Jeong Kim, Chan-Su Rha, Myoung Chong Song, Sungho Maeng, Ji-Ho Park, Dae-Ok Kim

**Affiliations:** 1Graduate School of Biotechnology, Kyung Hee University, Yongin 17104, Republic of Korea; 2Department of Gerontology (AgeTech-Service Convergence Major), Graduate School of East-West Medical Science, Kyung Hee University, Yongin 17104, Republic of Korea; 3Department of Food Science and Biotechnology, Kyung Hee University, Yongin 17104, Republic of Korea; 4AMOREPACIFIC R&I Center, Yongin 17074, Republic of Korea; 5Natural Products Research Institute, College of Pharmacy, Seoul National University, Seoul 08826, Republic of Korea

**Keywords:** acetylcholinesterase, Alzheimer’s disease, long-term potentiation, *N*-methyl-D-aspartate receptor, organotypic culture, pine bark

## Abstract

Alzheimer’s disease is the most common type of dementia with cognitive impairment. Various plant-derived phenolics are known to alleviate cognitive impairment in Alzheimer’s disease by radical scavenging and strengthening synaptic plasticity activities. Here, we examined the cognition-improving effect of *Pinus densiflora* Sieb. et Zucc. bark extract (PBE). We identified and quantified phenolics in the PBE using a UHPLC-Orbitrap mass spectrometer. To evaluate the cognition-enhancing effects of PBE, scopolamine-induced amnesic Sprague-Dawley (SD) rats (5 weeks old) and ion channel antagonist-induced organotypic hippocampal slices of SD rats (7 days old) were used. Twenty-three phenolics were tentatively identified in PBE, 10 of which were quantified. Oral administration of PBE to the scopolamine-induced SD rats improved cognitive impairment in behavioral tests. PBE-fed SD rats showed significantly improved antioxidant indices (superoxide dismutase and catalase activities, and malondialdehyde content) and reduced acetylcholinesterase activity in hippocampal lysate compared with the scopolamine group. PBE increased the long-term potentiation (LTP) induction and rescued LTP from blockades by the muscarinic cholinergic receptor antagonist (scopolamine) and *N*-methyl-D-aspartate channel antagonist (2-amino-5-phosphonovaleric acid) in the organotypic hippocampal slices. These results suggest that polyphenol-rich PBE is applicable as a cognition-improving agent due to its antioxidant properties and enhancement of LTP induction.

## 1. Introduction

Pine trees are conifers belonging to the genus *Pinus* of the family Pinaceae and are used for various purposes; they are economically important and widely distributed worldwide. *Pinus densiflora* Sieb. et Zucc. is a pine tree native to Korea, Japan, and Russia; it is also known as red pine because of its red bark [1]. The bark of *P. densiflora* is known to be rich in condensed tannin and its building blocks catechin and epicatechin, which have strong antioxidant capacity due to free hydroxyl (−OH) groups in their catechol groups [2,3]. Taxifolin (dihydroquercetin), which is commonly detected in high concentration in the bark of *Pinus* species, is also considered a potent antioxidant [1]. Due to various bioactive compounds, *P. densiflora* bark has beneficial health effects including antioxidant capacity [1], anti-obesity effects [4], and anti-hypertension activity [5].

Alzheimer’s disease (AD) is the most common form of dementia, accounting for more than 60% of all cases, and it is characterized by decreased memory and cognitive ability [6,7]. AD is characterized by neurofibrillary tangle deposition and amyloid ꞵ aggregation, which is influenced by hyperphosphorylation of tau. AD brains show pathogenic changes that make AD worse, including overwhelming reactive oxygen species (ROS) formation, synaptotoxicity [8], and disturbance of neurotransmissions [6]. Because the content of antioxidant enzymes in the brain is relatively low and the brain uses a lot of oxygen for oxidative metabolism, producing more ROS makes the brain more vulnerable to neuronal diseases like AD [9]. With work on the pathophysiology of AD-related ROS, research on plant extracts has been of interest as a potential ingredient for preventing or treating AD due to antioxidant phenolics, which could show radical-scavenging activity in the brain [10,11].

Long-term potentiation (LTP) is a crucial memory-formation mechanism in the neurons of the brain that increases synaptic strength [12]. Learning deficiencies and memory loss is commonly observed in LTP-impaired animals. Also, patients with AD showed synaptic dysfunction accompanied by loss of LTP. It has been clearly demonstrated that oxidative stress status is strongly related to LTP impairment [11,13]. Thus, studies on the synaptic strengthening effect of plant-derived antioxidant phenolics on AD models have been reported based on the LTP mechanism in the hippocampus [11,13].

The phenolic phytochemical profiling of *P. densiflora* Sieb. et Zucc. bark was limited to several major compounds, which have also been reported in another *Pinus* genus [14,15]. Also, the cognition-enhancing effects of PBE in the AD animal model and its mechanism that leads to the improvement of cognitive function are currently not well understood. As far as we know, no studies on the LTP-enhancing effect in *ex vivo* AD hippocampal slice model have been performed for investigating the potential of PBE. Therefore, we used a high-resolution ultra-high performance liquid chromatography (UHPLC)-Orbitrap mass spectrometer (MS) to identify phenolic compounds, including previously unreported phenolic compounds in *P. densiflora*, and to quantify several phenolic compounds in the PBE. We evaluated whether phenolic-rich PBE would prevent the decline of memory and learning function in 6-week-old male Sprague-Dawley (SD) rats with scopolamine (SCOP)-induced learning and memory deficits. Furthermore, LTP-enhancing effects of PBE that could be related to PBE-mediated recovery of memory and learning function were estimated in organotypic hippocampal slices from the brain of 7-day-old SD rats. The effects of PBE on ion channels involved in LTP induction using ion channel antagonists were further elucidated in this study. 

## 2. Materials and Methods

### 2.1. Chemicals

Hot water-extracted PBE was obtained from Nutrapharm Ltd. (Yongin, Republic of Korea). SCOP hydrobromide, protocatechuic acid, (+)-catechin, procyanidin B1, procyanidin B2, procyanidin B3, taxifolin, caffeic acid, quercetin, (−)-epicatechin, Hank’s Balanced Salts, 4-(2-hydroxyethyl)piperazine-1-ethanesulfonic acid (HEPES), 2-thiobarbituric acid, trichloroacetic acid, 1,1,3,3-tetramethoxypropane, acetylthiocholine, 6-cyano-7-nitroquinoxaline-2,3-dione (CNQX), DL-2-amino-5-phosphonovaleric acid (DL-AP5), malondialdehyde tetrabutylammonium salt, and 5,5′-dithio-bis-(2-nitrobenzoic acid) were purchased from Sigma-Aldrich Co., LLC (St. Louis, MO, USA). Lysis buffer was obtained from Noble Bio, Inc. (Hwaseong, Republic of Korea). Protease inhibitor cocktail was purchased from GenDEPOT (Barker, TX, USA). Superoxide dismutase (SOD) assay kit (DG-SOD400) and catalase assay kit (DG-CAT400) were purchased from DoGenBio Co., Ltd. (Seoul, Republic of Korea). Horse serum and penicillin-streptomycin were purchased from Biowest (Nuaillé, France) and Gibco BRL (Grand Island, NY, USA), respectively. LC-MS grade water and acetonitrile were purchased from Merck (Darmstadt, Germany).

### 2.2. Identification of Phenolic Compounds in PBE Using UHPLC-Orbitrap MS/MS

For UHPLC-Orbitrap-MS/MS analysis, PBE was solubilized in 30% (*v*/*v*) aqueous methanol and sonicated for 10 min, then the mixture was filtered using a 0.22-μm polytetrafluoroethylene membrane filter (Merck). The UHPLC-Orbitrap-MS/MS analysis of PBE phenolic compounds was performed using a Vanquish Horizon UHPLC system coupled to Orbitrap Exploris 120 MS (Thermo Fisher Scientific Inc., Waltham, MA, USA). Separation was performed on a Hypersil GOLD™ Vanquish C18 UHPLC column (2.1 mm × 150 mm, 1.9 μm; Thermo Fisher Scientific Inc., Waltham, MA, USA) equipped with a Hypersil GOLD™ Vanquish C18 with guard column (Thermo Fisher Scientific Inc.) at 45 °C with a sample injection volume of 1 μL. The two mobile phases were 0.1% (*v*/*v*) formic acid in water (solvent A) and acetonitrile (solvent B) at a flow rate of 0.3 mL/min. The gradient used was as follows: 95% A/5% B at 0–2 min, 85% A/15% B at 7 min, 75% A/25% B at 20–23 min, 5% A/95% B at 33 min, and 95% A/5% B at 38–40 min. The MS measurement was done in negative electrospray ionization (ESI) mode with a voltage 2.5 kV acquiring in full−MS/top 3 data-dependent MS/MS spectra in the mass range of 50–1500 Da, and ESI methods were as follows: vaporizer temperature at 320 °C, ion transfer tube temperature at 320 °C, sheath gas (N_2_) at 50 arb, and auxiliary gas (N_2_) at 10 arb. The higher energy collision dissociation was set to assisted collision mode with normalized collision energies of 30% and 40%. The data-dependent MS/MS were performed on the most intense ions detected in the full scan MS, with dynamic exclusion method which allowed us to avoid repeating MS^2^ (Figure 1A). 

### 2.3. Quantification and the Quality Parameter of Phenolic Compounds in PBE Using UHPLC-DAD

The quantification of phenolic compounds in PBE was performed using a Vanquish Horizon UHPLC system coupled to a Vanquish diode array detector (DAD; Thermo Fisher Scientific Inc.). Individual phenolic compounds in PBE were quantified at various UV-visible wavelengths using nine reliable standard phenolic compounds: protocatechuic acid (260 nm), (+)-catechin, (−)-epicatechin, procyanidin B1, procyanidin B2 and procyanidin B3 (280 nm), taxifolin and taxifolin isomer (290 nm), caffeic acid (320 nm), and quercetin (380 nm). The quality parameter of UHPLC-DAD quantification was determined using the serial dilution of the mixture of nine standard solutions, the calibration curve, correlation coefficient (*R*), limit of detection (LOD), and limit of quantification (LOQ). The calibration curves were calculated using standard solutions at six different concentrations injected three times each.

### 2.4. Animals

Thirty-two 5-week-old male SD rats (approximately 200 g) were purchased from Saeron Bio Inc. (Uiwang, Republic of Korea). Rats were housed in a laboratory cage under a controlled temperature of 23 °C, a relative humidity of 56%, and a 12 h light-dark cycle from 8:00 a.m. to 8:00 p.m. Rats had access to a standard diet (5L79; Orient Bio Inc.; Seongnam, Republic of Korea) and water *ad libitum* throughout the experiment period. All animal procedures complied with the Institutional Care and Use Committee of Kyung Hee University with approval number: KHUASP(SE)-18-040 (approval date: 12 January 2019) and were performed following the guiding principles for the care and use of animals approved by the Council of the National Institutes of Health Guide for the Care and Use of Laboratory Animals.

### 2.5. Oral Administration of PBE

After a week of adaptation, rats were randomly assigned to a control group, the SCOP group, and two PBE-treated groups. Each group consisted of eight rats. Tap water (control group and SCOP group) or PBE (15 and 30 mg/kg body weight (BW)/day in PBE15 and PBE30 groups, respectively) were orally administered to rats at 9 a.m. after the adaptation. Behavioral experiments were performed after 12 days of pre-administration of tap water or PBE. During the experimental period, tap water or PBE was orally administered an hour before the experiment, and SCOP was injected intraperitoneally into SCOP and PBE groups 30 min before behavioral tests (Figure 1B).

### 2.6. Behavioral Experiments

#### 2.6.1. Y-maze Test

The Y-maze test is a cognition experiment that estimates short-term memory and spatial learning capacity of animals. The experimental equipment consisted of a Y-shaped maze and a camera on the ceiling above the maze to capture the movement of the rats. The maze had the same angle (120°) for the three arms, and each arm had length × height × width of 45 cm × 35 cm × 10 cm. Each arm was randomly assigned as A, B, and C. The position and the movement of the rats in the maze were recorded with a camera for 10 min. The entry and exit of each arm were recorded when the rat’s hind leg crossed half of the arm. When the rat alternated between the three arms, the alternation [e.g., a pattern of ABCABAB has three alternations (ABC, BCA, and CAB)] was measured and incorporated in the calculation for the alternation score percentage (%) defined as the following equation: $$\mathrm{Alternation}\;\left(\%\right)=\frac{\mathrm{Number}\;\mathrm{of}\;\mathrm{spontaenous}\;\mathrm{alternations}}{\left(\mathrm{Total}\;\mathrm{number}\;\mathrm{of}\;\mathrm{arm}\;\mathrm{entries}\;-2\right)\;}\times 100$$

#### 2.6.2. Step-through Passive Avoidance Test

The step-through passive avoidance test is a fear-motivated test that evaluates the long-term avoidance memory ability of animals. The experimental apparatus consisted of a dark chamber separated by an automatic guillotine door and a steel rod on the bottom that gave an electric shock to rats. Experimental rats underwent two separate trials: a training trial for the acquisition of fear and a retention trial to estimate whether the memory of the fear remains. In the training trial, a rat was placed in a bright chamber. After 20 s of adaptation time, the entrance to the dark chamber opened with a little noise. When the rat moved into the dark room, the middle partition closed mildly, and then a 0.5 mA of electric foot shock was delivered to the rat for 5 s through the steel rod on the bottom. After the training trials, rats returned to the cage and received a retention trial to measure their memory abilities after 24 h. In the retention trial, the rat moved to the bright chamber, and after 20 s, the passage to the dark room opened with a sound. The time of step-through latency of the rat was measured for 300 s. 

#### 2.6.3. Morris Water Maze Test

The Morris water maze test is a behavioral experiment used to evaluate long-term spatial learning and memory ability of rats as they swim in a water tank. The experimental equipment consisted of a steel circular water tank with a diameter of 180 cm and a height of 45 cm and a platform inside the water tank. The water tank was filled with water (23 ± 1 °C) slightly above the platform height, making it difficult to distinguish the platform from the water. A rat took four days of training consisting of four training sessions. In each, the rat was placed in different quadrants of the water tank and allowed to swim freely for 1 min. If the rat was found on the platform, it was returned to the cage 10 s after reaching the platform. If the rat did not find the platform, the rat was guided to the platform through the indicator rod and allowed to stay on the platform for 10 s. After four days of training, the platform was eliminated, and the rats were placed in the test region rather than where they were placed during the training period, and the rats swam freely for 90 s. The movement of the rats inside the water tank was recorded with a camera mounted on the ceiling above the water tank, and the path and time of the rats were analyzed using the SMART video tracking system (SMART v3.0; Panlab, Barcelona, Spain).

### 2.7. Biochemical Experiments

#### 2.7.1. Collection of Rat Hippocampus

After the three behavioral tests, animals were anesthetized with isoflurane, and then the hippocampus was rapidly harvested. The hippocampus was homogenized with lysis buffer. The homogenized tissues in the lysis buffer were sonicated (NRE-02; Next Advance, Troy, NY, USA) and centrifuged at 18,403× *g* for 20 min at 4 °C using a centrifuge (PK121R; ALC International S.R.L., Cologno Monzese, Italy). The supernatant of tissue lysate was collected and stored at −80 °C prior to analysis.

#### 2.7.2. Measurement of SOD and Catalase Activity

SOD and catalase activities of the hippocampus lysate were measured according to the manufacturer’s instructions using a microplate reader (Infinite M200; Tecan Group Ltd., Männedorf, Switzerland). SOD activity of the hippocampal lysate was expressed as the inhibition rate of the blank without sample input. Catalase activity of the hippocampal lysate was determined using a catalase standard calibration curve.

#### 2.7.3. Measurement of Total Lipid Oxidation

The determination of lipid peroxidation of the hippocampus was measured according to the modified method of Draper et al. [16] Briefly, 100 μL of appropriately diluted hippocampus supernatant was mixed with 200 μL of 10% (*v*/*v*) trichloroacetic acid, and then the mixture was incubated in ice for 10 min to precipitate the protein. The acidified hippocampus supernatant was centrifuged at 18,403× *g* for 5 min and moved to other microtubes. Then, 200 μL of the supernatant or 1,1,3,3-tetramethoxypropane standard was mixed with 200 μL of 0.67% (*w*/*v*) 2-thiobarbituric acid in the microtubes, and the mixture was heated for 10 min at 100 °C. After the heat reaction, the reactant was cooled, moved to a 96-well plate, and measured at 531 nm using a microplate reader (Infinite M200; Tecan Group Ltd.). MDA content of tissue lysate of plasma (nmol MDA/mg protein of hippocampus lysate) was determined using 1,1,3,3-tetramethoxypropane standard calibration curve.

#### 2.7.4. Measurement of AChE Activity

The determination of AChE activity of the hippocampus was measured according to the modified method of Ho and Ellman [17]. Acetylthiocholine iodide was used as a substrate for the measurement of AChE activity. Five microliters of appropriately diluted hippocampus lysate and 65 μL of 50 mM sodium phosphate buffer (pH 7.4) were incubated for 15 min at 37 °C. After incubation, 70 μL of Ellman’s reaction mixture [0.5 mM acetylthiocholine and 1 mM 5,5′-dithio-bis(2-nitrobenzoic acid) in a 50 mM sodium phosphate buffer (pH 7.4)] was added and incubated for 10 min at 37 °C. The mixture was then measured at 415 nm using the microplate reader (Infinite M200; Tecan Group Ltd.). AChE activity was expressed in a relative percentage of the control group (100%).

### 2.8. Electrophysiological Experiments Using Organotypic Hippocampal Slice Culture

#### 2.8.1. Organotypic Hippocampal Slice Culture

Organotypic hippocampal slices were prepared according to the previous method developed by Stoppini et al. [18] using 7-day-old male SD rats. Briefly, the rat hippocampus was harvested and sliced transversely at 350 μm thick using a tissue chopper (Mickle Laboratory Engineering Co., Surrey, UK). Then, five to six hippocampus slices were plated onto a 0.4 μm membrane insert (Millicell-CM; Merck Millipore, Bedford, MA, USA) and incubated at 36 °C with 5% CO_2_ and a culture medium (minimum essential medium supplemented with 20 mM HEPES, 6 g/L D-glucose, 1 mM L-glutamine, 25% Hank’s balanced salts, 25% horse serum, and 1% penicillin-streptomycin; pH 7.1). The culture medium was changed using a fresh one every two days, and slices, which had been cultured for 12–14 days, were used for the experiment.

#### 2.8.2. Preparation of Organotypic Hippocampal Slice Tissue on the Microelectrode Array (MEA) Probes

The cultured hippocampal slice was transferred from a membrane insert into an 8 × 8 MEA of 10 μm-diameter electrodes spaced 100 μm apart (Multi Channel Systems MCS GmbH, Reutlingen, Germany) pre-coated with 0.01% polyethylenimine. The slice was stabilized at 33°C with 95% O_2_ and 5% NaCl in artificial cerebrospinal fluid (aCSF; including of 114 mM NaCl, 26 mM NaHCO_3_, 10 mM D-glucose, 3 mM KCl, 2 mM CaCl_2_, 1 mM MgCl_2_, and 20 mM HEPES; pH 7.4) for 30 min. The array solution was grounded by Ag/AgCl pellets.

#### 2.8.3. Induction of LTP in Organotypic Hippocampal Slice Tissue

Experiments were performed using the MEA system (all from Multi Channel Systems MCS GmbH), which included a stimulator (STG1004), amplifier (MEA1060), temperature control unit, MEA, and data acquisition software (www.multichannelsystems.com; accessed on 3 May 2020). The Schaffer collateral and commissural pathways were selected based on morphological structure and empirical electrical responses. The bipolar electrical stimulation was applied to the stratum radiatum CA2 region to stimulate the Schaffer collateral and commissural pathways, and the amplifier was located in a faraday cage. The intensity of the bipolar test pulse baseline stimulation was set to 100 µA and optimized to yield 40–65% of the maximum tissue response. The theta-burst stimulation consisted of 300 biphasic pulses with three trains of pulses at 100 Hz for 1s (train interval of 5 min). Each experiment consisted of 30 min of a baseline recording of field excitatory postsynaptic potentials (fEPSPs) at one stimulation per min, 10 min of theta-burst stimulation, and 50 min of post high-frequency stimuli (HFS) fEPSP measurements. Schaffer collateral and commissural pathways were selected based on morphological structure and electrical stimulation response. The hippocampal slices for the LTP induction were continuously provided with aCSF at a rate of 3 mL/min. PBE (25, 50 and 100 mg/L)- or antagonist (300 μM SCOP, 50 μM DL-AP5, and 10 μM CNQX)-treated hippocampal slices were pretreated with drugs dissolved in aCSF at a rate of 3 mL/min after 10 min of baseline recording (Figure 1C).

#### 2.8.4. Electrophysiology Data Processing

Unfiltered data were sampled from 60 recording channels at 25 kHz using the Recorder-Rack software (Multi Channel Systems MCS GmbH). MC_Rack (version 3.2.1; Multi Channel Systems MCS GmbH) was used for digitizing the analog MEA signal and isolating fEPSPs from triggering amplitudes over 40 mV.

### 2.9. Statistical Analysis

All data were expressed as the mean ± standard error of the mean. Statistical analysis was performed using a SPSS software (Version 23.0; IBM SPSS Statistics Inc., Chicago, IL, USA). The significance of differences in average values was analyzed using a one-way analysis of variance, followed by a Tukey’s honestly significant difference test. The significant levels are represented as asterisks and hashtags (* *p* < 0.05, ** *p* < 0.01, *** *p* < 0.001 and ^#^
*p* < 0.05, ^##^
*p* < 0.01, ^###^
*p* < 0.001).

## 3. Results

### 3.1. Profiling and Determination of Phenolic Compounds Using UHPLC-Orbitrap MS

#### 3.1.1. Identification of Phenolic Compounds Using UHPLC-Orbitrap MS

Twenty-three phenolic compounds in PBE were tentatively identified by MS^2^ spectra and retention time using UHPLC-Orbitrap MS and reference data (Table 1). The twenty-three phenolic compounds were composed of six procyanidins (peaks 4, 6, 8, 10, 13, and 15), five phenolic acids (peaks 1, 3, 5, 9, and 12), one phenolic aldehyde (peak 2), nine flavonoids (7, 11, 16, 17, and 19–23), and two unknown phenolic compounds (peaks 14 and 18).

Peaks 4, 6, and 15 were associated with as procyanidin B-type dimer by comparison of MS^2^ fragments of reference standards and an in-house library. Peaks 4, 6, and 15 showed negative molecular [M − H]^−^ ions at *m*/*z* 577.1337 in common and the typical fragmentation patterns of procyanidin B-type observed by the negative ion mode of MS^2^, following heterocyclic ring fission (HRF), retro-Diels-Alder (RDA), and quinone methide (QM) reactions [19]. The first typical fragment ion at *m*/*z* 451.1010 (or 451.1009) was formed by the elimination of 1,3,5-trihydroxybenzene (THB) [M −H − 126]^−^ in the HRF reaction. The first typical fragment ion at *m*/*z* 451.1010 (or 451.1009) was formed by the elimination of THB [M − H − 126]^−^ in the HRF reaction. The molecular ion of the eliminated THB was clearly detected at *m*/*z* 125.0236. In the RDA reaction, the typical fragment ions at *m*/*z* 425.0853 and 407.0753 were generated from the elimination of hydroxyvinyl benzenediol (HVB), [M − H − 152]^−^, in a catechin moiety of the extension unit and further loss of a H_2_O, [M − H −152 − 18]^−^, respectively. The major fragment ion at *m*/*z* 289 [(epi)catechin − H]^−^ was produced from QM cleavage reaction in procyanidin B-type fragmentation, followed by loss of the CO_2_ (−44 amu) fragment ion at *m*/*z* 245.0806, and then loss of (CH_2_CO; −84 amu) fragment ions at *m*/*z* 161.0235 (Appendix A). Finally, peaks 4, 6, and 15 were identified as procyanidin B1, B3, and B2, respectively, by comparison with the retention time of reference standards.

Peaks 8, 10, and 13 were observed with molecular ions [M − H]^−^ at *m*/*z* 865.1957 and identified as procyanidin trimers (Appendix A). The diagnostic peak in the MS^2^ fragment ion was *m*/*z* at 577.1334 [M − H − (epi)catechin]^−^, which corresponds to procyanidin dimer. Typical fragment ions of procyanidin B-type dimer at *m*/*z* 451.1009 [M − H − (epi)catechin − THB]^−^, *m*/*z* 425.0840 [M − H − (epi)catechin − HVB]^−^, and *m*/*z* 289.0703 [(epi)catechin − H]^−^ were detected from HRF, RDA, and QM reactions, respectively. Unfortunately, it was not possible to confirm the exact structure of the procyanidin trimer of each of the four peaks. However, we confirmed that various procyanidin trimers were biosynthesized using procyanidin B1, B2, and B3 as mediators (Appendix A).

Peak 1 with the [M − H]^−^ ion at *m*/*z* 167.0344 was fragmented into MS^2^ fragment ions at *m*/*z* 149.0235 and at *m*/*z* 123.0444, corresponding to the loss of H_2_O (−18 amu) and CO_2_ (−44 amu), respectively. The negative molecular ion of peak 2 was detected at *m*/*z* 181.0500, and MS^2^ fragment ions were presented at *m*/*z* 151.0392, *m*/*z* 133.0287, and *m*/*z* 123.0444. Thus, peaks 1 and 2 were tentatively identified as hydroxymandelic acid and syringaldehyde, respectively, according to previous literature [20]. The precursor [M − H]^−^ ion of peak 4 was observed at *m*/*z* 153.0189, and the characteristic MS^2^ fragment ion was detected at *m*/*z* 109.0288 from the loss of CO_2_ (−44 amu). This peak was identified as a protocatechuic acid by comparison with a reference standard. 

Peaks 5 and 12 were detected as the same negative molecular ion [M − H]^−^ at *m*/*z* 337.0915. The MS^2^ fragment ions of peaks 5 and 12 were detected at *m*/*z* 191.0566 for [quinic acid − H]^−^, *m*/*z* 173.0449 for [M − H − *p*-coumaric acid]^−^ or [quinic acid − H − H_2_O]^−^, *m*/*z* 163.0394 for [*p*-coumaric acid − H]^−^, and *m*/*z* 119.0497 for [*p*-coumaric acid − H − CO_2_]^−^, indicating that these two compounds are *p*-coumaroylquinic acids (Appendix A). As previously reported in the literature [21], 3-*p*-coumaroylquinic acid showed higher intensity at *m*/*z* 163.0394, but 4-*p*-coumaroylquinic acid showed higher intensity at *m*/*z* 173.0449, which was generated by the elimination of water molecule from quinic acid ion (*m*/*z* 191.0566) (Appendix A). Thus, peaks 4 and 11 were tentatively identified as 3-*p*-coumaroylquinic acid and 4-*p*-coumaroylquinic acid, respectively. 

Peaks 7 and 11 with [M − H]^−^ ions at *m*/*z* 289.0706 were identified as (+)-catechin and (−)-epicatechin, respectively, by comparison with retention time, and MS^2^ fragments of reference standards and an in-house database. MS^2^ fragment ions of two peaks at *m*/*z* 245.0808, generated by loss of CO_2_ (−44 amu), and the loss an additional molecule of CH_3_CO (−42 amu) formed the fragment ion at *m*/*z* 203.0704. 

Peaks 17 and 23 with [M − H]^−^ ions at *m*/*z* 303.0502 and 301.0340 were identified as taxifolin and quercetin, respectively, based on comparisons with retention time and MS^2^ fragment ions of reference standard. The characteristic MS^2^ fragment ions of peaks 17 and 19 were observed at *m*/*z* 285.0392 for [M − H − H_2_O]^−^, and further fragment ions occurred at *m*/*z* 217.0496 after *ꞵ*-hydroxylation on ring A (−68 amu) and at *m*/*z* 175.0390 from the loss of CH_3_CO (−42 amu). Also, MS^2^ fragment ions of peak 23 were at *m*/*z* 273.0389 ([M − H − CO]^−^) and *m*/*z* 178.9976 from the fragmentation of B ring loss. Peak 20 with [M − H]^−^ ions at *m*/*z* 287.0549 had MS^2^ fragment ions at *m*/*z* 259.0599 and *m*/*z* 243.0651 corresponding to the loss of CO (−28 amu) and the loss of CO_2_ (−44 amu), respectively. Peak 20 was tentatively identified as dehydroxyltaxifolin. 

Peak 16 was observed with negative molecular ions at *m*/*z* 465.1020, and its MS^2^ fragment aglycone ions (Y_0_) and radical aglycone ions ([Y_0_ − H]^•−^) were detected at *m*/*z* 304.0529 and *m*/*z* 303.0494, respectively, by the elimination of glycoside (−162 amu) (Appendix A). To distinguish 3-*O*- and 7-*O*-monoglycoside of flavones and flavanonol, the relative abundance ratio of [Y_0_ − H]^•−^/Y_0_ was used as a diagnostic. The MS^2^ spectrum showed an intensity ratio of [Y_0_ − H]^•−^/Y_0_ (> 1) for 3-*O*-monoglycoside. Thus, peak 16 was assigned as taxifolin 3-*O*-glucoside. 

The precursors of peaks 21 and 22 in the negative mode were detected at *m*/*z* 447.0915 and *m*/*z* 463.0864, respectively. From the MS^2^ spectra of each peak, the major fragment ions of peak 21 were observed at *m*/*z* 301.0336 and *m*/*z* 300.0261 by the elimination of deoxyhexose (−146 amu) from molecular ions, whereas the major fragment ions of peak 22 were shown at *m*/*z* 301.0338 and *m*/*z* 300.0261 from the loss of hexose (−162 amu) from molecular ions. Both compounds appeared to be glycosyl flavonoids with different glycosides but the same aglycone. Following the same approach as for the identification of peak 16 (taxifolin 3-*O*-glucoside), peaks 21 and 22 were identified as quercetin 3-*O*-rhamnoside and quercetin 7-*O*-glucoside, respectively (Appendix A).

#### 3.1.2. Quantification of Phenolic Compounds Using UHPLC-Orbitrap MS

The concentrations, calibration curve, correlation coefficient, LOD, and LOQ of 10 phenolic compounds in PBE are presented in Table 2. Concentrations of these 10 major phenolics in PBE decreased as follows: procyanidin B1 > taxifolin > (+)-catechin > protocatechuic acid > procyanidin B3 > taxifolin isomer > procyanidin B2 > caffeic acid > (−)-epicatechin > quercetin. The quantification using reference standard showed *R* > 0.999, LOD ≤ 0.25 μg/mL, and LOQ ≤ 0.77 μg/mL that, in general, provides acceptable validation values.

### 3.2. In Vivo Effects of PBE on Memory Deficit in SCOP-Induced SD Rats

#### 3.2.1. PBE Improved SCOP-induced Memory-Cognitive Behavior Disorder

A short-term spatial memory test was conducted using the Y-maze test. Rodents have a curiosity to explore new environments rather than previously visited ones, which leads to alternative behavior in the Y-maze test. As shown in Figure 2A, the alternative behavior (72.8%) of the control group was higher compared with those of the other experimental groups. The alternative behavior (38.8%) of the SCOP-treated group was significantly lower than that of the control group. However, the PBE15 and PBE30 groups had alternative behaviors of 64.4% and 66.9%, respectively, which were higher than the SCOP group (Figure 2A). However, the total arm entry of all groups was not significantly different from one another, suggesting that the increase in alternative behavior in rats by PBE is not due to locomotor depression (Figure 2B).

Acquisition of avoidance memory was evaluated by a step-through passive avoidance test, which estimates the learning ability of rats to avoid unpleasant memories using the dark-loving habit of rats. In the training trial, rats in all groups moved to the dark room immediately (10.3 s) after the guillotine door opened and then received an electric foot shock (Figure 2C). In the retention trial, the control group well remembered the electric foot shock from the training period and showed the longest step-through latency time (214.8 s) in all experimental groups (Figure 2C). The SCOP group did not remember the electric foot shock of the training trial and immediately moved to the dark room as the training trial (Figure 2C). However, the PBE15 and PBE30 groups had significantly higher step-through latency times (respective 132.3 and 163.1 s) than the SCOP group (4.9 s) in the retention trial (Figure 2C).

Long-term spatial learning and memory were measured using the Morris water maze test. During the training period of 4 days, the control, PBE15, and PBE30 groups had shorter latency to the platform than the SCOP group (Figure 2D). In the retention trial, the control group had the highest target zone navigation (42.3%), while the SCOP group had significantly lower target zone navigation (29.4%) (Figure 2E). However, the PBE15 and PBE30 groups had a higher target zone navigation (37.0% and 38.8%, respectively) than the SCOP group (Figure 2E). 

#### 3.2.2. PBE Improved SCOP-induced Enzymatic Antioxidant Defense System Defect and Inhibited AChE Activity

In this study, the biochemical indicators (SOD, catalase, MDA, and AChE) related to cognitive and memory function in the brain were investigated using the hippocampus extracted from the SD rats after oral administration of PBE to SD rats for 20 days and three subsequent behavioral tests. Figure 3 shows the results of these biochemical indicators of hippocampus lysate from the SD rats orally administered PBE. SOD forms part of the enzymatic antioxidant defense system in the body and breaks down superoxide radical anions into hydrogen peroxide and water. As shown in Figure 3A, the control group showed the highest SOD activity (94.9%) in the hippocampus, while the SCOP group showed the lowest SOD activity (91.0%) in the hippocampus. However, the PBE15 and PBE30 groups had significantly higher SOD activities (94.5% and 95.1% of the control group, respectively) in the hippocampus than the SCOP group (Figure 3A).

Catalase is a critical enzyme of the antioxidant defense system in the body that degrades hydrogen peroxide to water and oxygen. The control group showed the highest catalase activity (36.1 mmol/min/mg protein) in the hippocampus, and the SCOP group showed significantly lower catalase activity (20.7 mmol/min/mg protein) in the hippocampus than the control group (Figure 3B). However, the PBE15 and PBE30 groups had significantly higher catalase activity (32.8 and 30.6 mmol/min/mg/protein, respectively) in the hippocampus than the SCOP group (Figure 3B).

MDA is a product of unsaturated fatty acid peroxidation and is used as a biomarker for oxidative stress of brain cells. As shown in Figure 3C, the MDA content in the hippocampus of the control group was 0.74 mmol/mg protein, and the SCOP group showed a higher MDA content (0.81 mmol/mg protein) than the control group. However, the PBE15 and PBE30 groups had significantly lower MDA content (0.50 and 0.56 mmol/mg protein, respectively) in the hippocampus than the SCOP group (Figure 3C).

In Figure 3D, AChE activity (178.1% of the control group) in the hippocampus of the SCOP group was significantly higher than that of the control group (100%). However, the PBE15 and PBE30 groups showed lower AChE activities (100.3% and 124.1% of the control group, respectively) in the hippocampus than the SCOP group (Figure 3D).

### 3.3. Effects of PBE on Long-term Synaptic Plasticity in Organotypic Hippocampal Slices

#### PBE Enhanced LTP and Rescued LTP Induction Failure by Synaptic Channel Antagonists in CA1 Hippocampal Region

Three concentrations of PBE (25, 50, and 100 mg/L) dissolved in aCSF were continuously fed to the hippocampal slices to determine concentrations effective for LTP induction. LTP induction after HFS increased in the hippocampal slices treated with PBE at 25 and 50 mg/L compared to the control group, while treatment of the hippocampal slices with PBE at 100 mg/L did not increase LTP from that of the control group (Figure 4A). In addition, treatment of hippocampal slices with PBE at 25 and 50 mg/L showed 163.6% and 189.3% of fEPSP total activity for 30–40 min after HFS (1.1- and 1.2-fold of the control group, respectively), but there was a significant difference between the control group and the group treated with 50 mg/L of PBE (Figure 4B). On the other hand, PBE treatment of 100 mg/L showed lower fEPSP (152.2% of fEPSP total activity) than the control group (Figure 4B). Hence, PBE treatment at 50 mg/L was used in subsequent experiments.

As shown in Figure 4C, SCOP treatment to hippocampal slices decreased LTP induction by HFS and fEPSP total activity. However, co-treatment with 50 mg/L of PBE and SCOP (i.e., PBE50+SCOP group) resulted in a recovery of LTP compared to the SCOP group (Figure 4C). As shown in Figure 4D, the fEPSP total activity of 30–40 min after HFS of SCOP treatment was 121.1% (78.4% of the control group), but co-treatment with PBE and SCOP recovered fEPSP total activity up to 141.2% (91.1% of the control group).

The *N*-methyl-D-aspartate (NMDA) receptor antagonist (DL-AP5; 50 μM) and α-amino-3-hydroxy-5-methyl-4-isoxazolepropionic acid (AMPA) receptor antagonist (CNQX; 50 μM) were added to hippocampal slices to elucidate the mechanism by which PBE improved LTP induction (Figure 4E–H). DL-AP5 reduced HFS-induced LTP, and co-treatment with both PBE and DL-AP5 resulted in a recovery of LTP compared to the group treated with DL-AP5 alone (Figure 4E). As shown in Figure 4F, DL-AP5 treatment had 91.7% (60.0% of the control group) of fEPSP total activity at 30–40 min after HFS, but co-treatment with PBE and DL-AP5 significantly recovered fEPSP total activity up to 126.4% (80.7% of the control group). However, the CNQX reduced HFS-induced LTP, and co-treatment with PBE and CNQX showed no recovery (Figure 4G,H). The fEPSP total activity after HFS of CNQX treatment was 98.4% (64.2% of the control group), and co-treatment with PBE and CNQX was 103.7% (67% of the control group) (Figure 4H). There was no significant difference between the CNQX treatment only and co-treatment with PBE and CNQX in LTP induction by HFS (Figure 4H).

## 4. Discussion

PBE has been reported to have antioxidant, anti-obesity, and anti-hypertension effects, but information on its phytochemical profiles is limited [1,4,5]. In this study, phenolic compounds contributing to the anti-amnesic effect of PBE in in vivo and ex vivo models were identified and quantified using UHPLC-Orbitrap-MS. A total of 23 phenolic compounds were identified, including 9 of them quantified as reference substances. Also, hydroxymandelic acid, syringaldehyde, 3-*p*-coumaroylquinic acid, 4-*p*-coumaroylquinic acid, quercetin 3-*O*-rhamnoside (quercitrin), and quercetin 7-*O*-glucoside (quercimeritrin) in *P. densiflora* were first identified in this study. 

Quantification showed that protocatechuic acid, taxifolin, and procyanidin B dimer and its building block (+)-catechin were the major compounds of PBE that occupied 98.2% of the total amount of quantified phenolic compounds (Table 2.). Notably, procyanidins and catechins were abundant in PBE. A previous study showed that the total amounts of procyanidins were 56 mg/g for *P. brutia*, 64.5 mg/g for *P. pinea*, 24.4 mg/g for *P. sylevestris*, and 40.7 mg/g for *P. nigra* [14], suggesting that procyanidin and its derivatives are the most abundant phenolic compounds present in the bark of the *Pinus* species. In our study, only three procyanidin dimers were quantified in PBE (sum of two procyanidin dimers; 17.95 mg/g). Given the existence of the other procyanidin dimers and trimers which were not quantified, the content of procyanidins in PBE may be similar or greater than in other species (Table 2). (+)-Catechin concentration was 10.08 mg/g in PBE, which showed similar or higher amounts than the bark of other *Pinus* species, 11.4 mg/g for *P. radiata* [22], 19.9 mg/g for *P. patula* [23], 8.4 mg/g for *P. sylvestris*, and 0.6 mg/g for *P. nigra* [24]. 

SCOP induces temporary cognitive memory impairment in rodents, such as rats and mice, thus it is used for cognitive behavior experiments, such as the Y-maze test, passive avoidance tests, and Morris water maze tests [10,25]. Previous studies reported that plant extracts containing antioxidant phenolic compounds could alleviate SCOP-induced memory-cognitive behavioral impairment in rodents [10,11,26,27]. In this study, we observed that SCOP-treated SD rats had cognitive impairment, whereas phenolic-rich PBE improved cognitive impairment in the three behavioral tests (Y-maze, passive avoidance, and Morris water maze tests) (Figure 2).

The brain is more susceptible to oxidative stress than other organs due to its high consumption of oxygen, high amounts of polyunsaturated fatty acids, fewer antioxidant defense systems, and neurotransmitter auto-oxidation. Persistent neuronal oxidative damage in the brain eventually advances to various neurodegenerative diseases, such as AD [28]. Thus, ROS detoxification to prevent or treat ROS-induced diseases in the brain by antioxidant enzymes, such as SOD, catalase, and glutathione peroxidase, is more important than in any other organs [28]. Acetylcholine (ACh) signaling, such as regulating AChE levels in the brain, is also an important mechanism involved in neuronal diseases because it plays an important role in cognitive performance, learning and memory processes [6,29]. In addition, decreased choline acetyltransferase activity, reduced ACh release, and increased AChE activity have been reported in patients with AD [6,30]. The SCOP-induced cognitive deficit models in an in vivo study have been reported to reduce antioxidant enzyme activity and exhibit cholinergic dysfunction, like increased AChE and decreased ACh as in AD patients [25].

Consistent with previous studies [11,25], the SCOP group showed significantly lower antioxidant enzyme activities, increased MDA content, and higher AChE activity. However, PBE treatment significantly increased SOD and catalase activities, decreased MDA content, and reduced AChE activity compared to the SCOP group (Figure 3). Many phenolic compounds are known to be effective in preventing the reduction of antioxidant enzyme activity and inhibiting AChE activity [10,25,26]. PBE and its phenolic compounds, such as catechin and taxifolin, were previously reported to reduce AChE activity and to have antioxidant capacities for scavenging ROS [1]. Therefore, antioxidants and AChE-inhibiting properties of PBE may contribute to cognition-enhancing effects in SCOP-induced cognitively impaired SD rats.

The impairment of memory and learning ability due to AD is closely related to LTP induction failure in the hippocampus. LTP induction is related to several ion channels such as the NMDA receptor and AMPA receptor. Therefore, we further determined whether PBE could recover the LTP induction in the SCOP-induced amnesic *ex vivo* model and elucidated which ion channels are involved in the mechanism of PBE action. We used 7-day-old SD rats, which differ from rats used in behavioral tests, since adult hippocampal slices have less resistance to physical damage than young hippocampal slices. Specifically, adult hippocampal nerve cells do not tend to survive when sliced [31]. Although some studies attempted to culture adult hippocampal slices, cell damage was inevitable [32,33]. Therefore, in our study, young hippocampal slices with excellent neuroplasticity from 7-day-old SD rats were used to withstand relatively long-term culture *ex vivo* and to circumvent neuronal cell loss.

The most effective concentration to increase LTP induction is 50 mg/L (Figure 4A,B). PBE, a complex extract containing various bioactive phenolic compounds, may affect numerous neuronal mechanisms, including γ-aminobutyric acid (GABA) receptor modulation. Taxifolin, one of the major bioactive compounds in PBE [5], has been reported to inhibit the increase in cytosolic Ca^2+^ concentration in GABAnergic neurons and decrease neuronal excitation [34]. The reason why high concentration of PBE (100 mg/L) did not increase the LTP induction is partly due to compounds like taxifolin, which play a role in inducing inhibitory postsynaptic potential (Figure 4A,B).

In the behavioral tests, the effects of PBE on improving cognitive and memory in the SCOP-induced cognitive impairment rat model were demonstrated (Figure 2). In addition, antioxidant biomarkers (SOD and catalase activities, and MDA content) and cholinergic function were improved in the hippocampal homogenate of rats fed with PBE (Figure 3). As an extension of these results, we determined that PBE improved the LTP disturbance in organotypic culture of the hippocampal slices induced using muscarinic acetylcholine receptor antagonist (SCOP; 300 μM). 

DL-AP5 inhibited LTP induction in the CA1 region of the hippocampal slices (Figure 4E,F). Co-treatment of hippocampal slices with PBE and DL-AP5 did not restore fEPSP total activity compared to the control, but co-treatment with DL-AP5 and PBE reversed LTP inhibition caused by DL-AP5 treatment and had 121% of fEPSP total activity (Figure 4E,F). DL-AP5 acts as a competitive NMDA antagonist and is known to reversibly block the NMDA receptor [35]. If the bioactive phenolic compounds in PBE are bonded to the allosteric or glycine binding sites of the NMDA receptor, co-treatment with DL-AP5 and PBE will restore the fEPSP level to a normal level. Our results imply that PBE reversed the LTP inhibition induced by the NMDA receptor antagonist by competing with DL-AP5. On the other hand, we observed that LTP was inhibited through the AMPA receptor antagonist CNQX and that the co-treatment with CNQX and PBE could not restore LTP inhibition (Figure 4G,H). These results suggest that LTP induction by PBE is attributed to NMDA receptor activation instead of AMPA receptor activation.

The results of this study consistently demonstrated the cognition-improving effects of PBE based on in vivo behavioral tests and LTP assessment in organotypic hippocampal slices. However, the differences in cognitive improvement effects of behavioral tests and biochemical indices in the rat hippocampus between 15 and 30 mg/kg BW/day dosages were not completely consistent. Also, the increase in LTP induction by PBE in the hippocampal slices did not show a dose-dependent trend in this study. Many types of phenolic compounds (such as procyanidins, catechin, and quercetin) having different molecular properties and bioavailability are found in PBE [36,37,38]. For example, pine bark contains a large number of procyanidin derivatives that have different bioavailability and bioactivity depending on the degree of polymerization [37]. Thus, considering the wide range of bioavailability of phenolic compounds in PBE in ex vivo and in vivo conditions, it is difficult to explain why no dose-dependent results were obtained in this study. Another reason for some of the discrepancies in the results of this study between in vivo behavioral tests and ex vivo LTP measurements may be due to differences in the ages of the animals used. We used 6-week-old (after adaptation period) SD rats for in vivo behavioral tasks but used 7-day-old SD rats for the ex vivo LTP study. 

There are some limitations in this study due to the difficulty of using dual animal models and the complexity of pharmacological effects by the use of extracts. However, our study determined the anti-amnesic effect of PBE in the cognitive impairment model, which validated in both ex vivo and in vivo models and provided a comprehensive phytochemical understanding of PBE. Further studies are needed to elucidate the clearer mechanism for the cognition-improving effects of PBE and its major bioactive compounds.

## 5. Conclusions

In conclusion, we presented the phenolic profiling of PBE and investigated the cognition-enhancing effects of PBE through animal behavior and electrophysiology tests. Twenty-three phenolic compounds were identified using UHPLC-DAD-Orbitrap MS, including compounds not previously found in the bark of *P. densiflora*. Oral administration of phenolic-rich PBE for three weeks showed cognition-enhancing effects on SCOP-induced amnesic rats in a Y-maze, passive-avoidance, and Morris water maze tests. PBE normalized activities of endogenous antioxidant enzymes including SOD and catalase and AChE activity in hippocampus of the rats. Furthermore, PBE promoted LTP formation and restored LTP inhibition by NMDA receptor antagonist DL-AP5 in organotypic cultured hippocampal slices. Our results provide comprehensive information on pharmaceutical and nutraceutical candidates for the prevention of cognitive disorders.

## Figures and Tables

**Figure 1 antioxidants-11-02497-f001:**
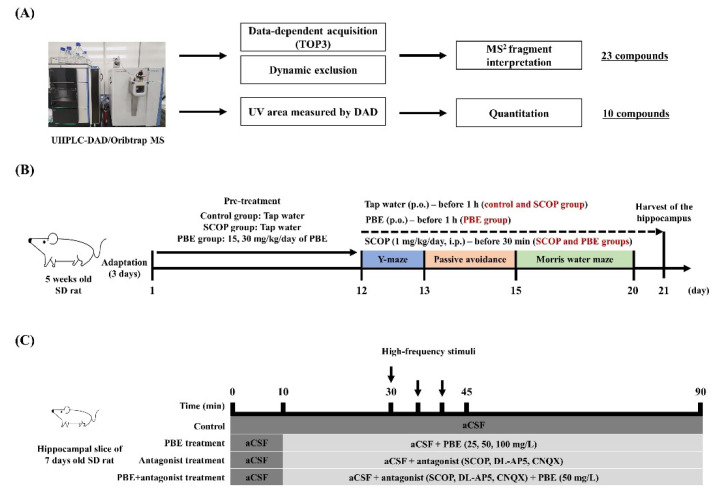
Experimental design in this study. (**A**) Phenolic analysis using UHPLC-DAD/Orbitrap MS/MS analysis, (**B**) Animal behavioral study in Sprague-Dawley (SD) rat and (**C**) measurement of long-term potentiation in hippocampal tissue of young SD rat. aCSF, artificial cerebrospinal fluid; CNQX, 6-cyano-7-nitroquinoxaline-2,3-dione; DAD, diode array detector; DL-AP5, DL-2-amino-5-phosphonopentanoic acid; MS, mass spectrometer; PBE, *Pinus densiflora* Sieb. et Zucc. bark extract; SCOP, Scopolamine; UV, ultraviolet.

**Figure 2 antioxidants-11-02497-f002:**
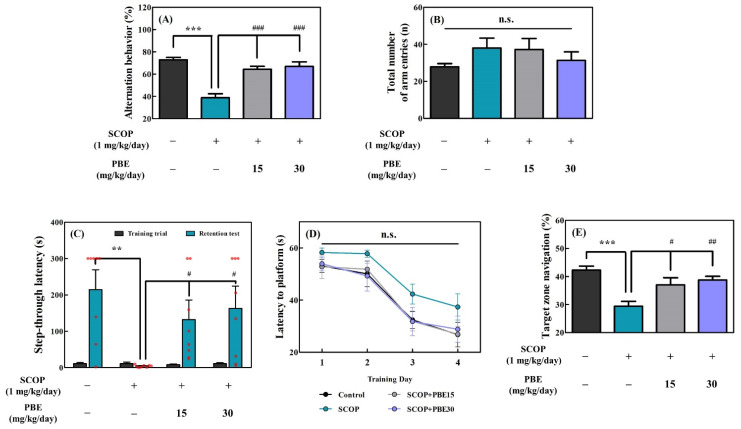
Effects of PBE on the animal behavior tests. SCOP (1 mg/kg/day body weight (BW); *i.p.*) was administered to SD rats 30 min before the tests (n = 8). PBE (15 and 30 mg/kg BW/day; *p.o.*) was administered to the PBE groups 60 min before the tests. (**A**) Alternation behavior and (**B**) total number of arm entries using the Y-maze test, (**C**) step-through latency time of acquisition trial and retention test in the passive avoidance test, and (**D**) latency time to platform in training days and (**E**) target zone navigation of acquisition trial using the Morris water maze test. Tukey’s honestly significant difference test: ** *p* < 0.01, *** *p* < 0.001 vs. control and ^#^
*p* < 0.05, ^##^
*p* < 0.01, ^###^
*p* < 0.001 vs. SCOP.

**Figure 3 antioxidants-11-02497-f003:**
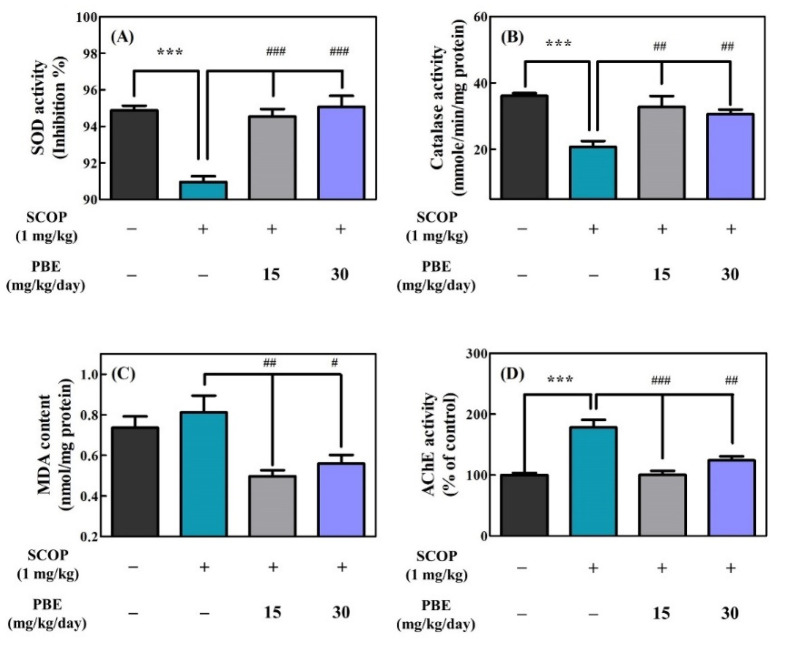
Effects of PBE on (**A**) superoxide dismutase (SOD) activity, (**B**) catalase activity, (**C**) malondialdehyde (MDA) content, and (**D**) acetylcholinesterase (AChE) activity in hippocampus lysate of SD rats (n = 8) after the behavioral tests. Tukey’s honestly significant difference test: *** *p* < 0.001 vs. control and ^#^
*p* < 0.05, ^##^
*p* < 0.01, ^###^
*p* < 0.001 vs. SCOP.

**Figure 4 antioxidants-11-02497-f004:**
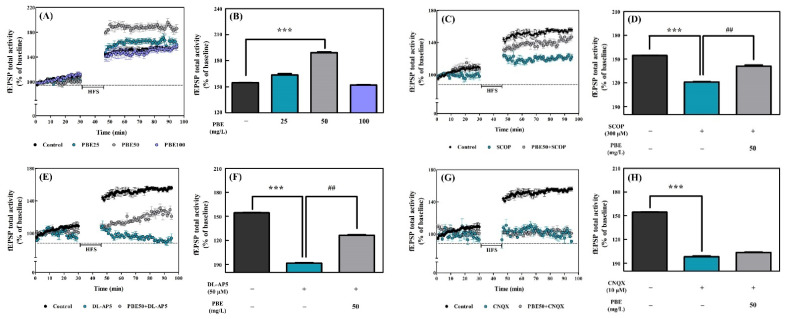
Effects of PBE, SCOP, DL-AP5, and CNQX on long-term potentiation (LTP) in organotypic hippocampal tissue from 7-day-old male SD rats (n = 6). (**A**) Time course of LTP from all recordings in control and PBE (25, 50, and 100 mg/L) groups, (**C**) time course of LTP in control, SCOP (300 μM), and SCOP with 50 mg/L of PBE, (**E**) time course of LTP in the control, DL-AP5 (50 μM) and DL-AP5 with PBE50, and (**G**) time course of LTP in the control, CNQX (10 μM) and CNQX with PBE50. (**B**,**D**,**F**,**H**) Average LTP amplitude measured at 30–40 min after high-frequency stimuli (HFS). fEPSP, field excitatory postsynaptic potential. Tukey’s honestly significant difference test: *** *p* < 0.001 vs. control and ^##^
*p* < 0.01 vs. SCOP.

**Table 1 antioxidants-11-02497-t001:** Phenolic compounds identified in *Pinus densiflora* Sieb. et Zucc. bark extract (PBE) using a high-resolution UHPLC-Orbitrap mass spectrometer in negative ionized mode.

Peak	Retention Time (min)	Proposed Compound	Molecular Formula [M − H]^−^	Calculated Mass [M − H]^−^	Measured Mass [M − H]^−^	Δ ppm	MS^2^ Fragments (Relative Abundance, %)
1	2.22	Hydroxymandelic acid	C_8_H_7_O_4_^−^	167.0345	167.0344	−0.60	149.0235 (11), 139.0393 (26), 123.0444 (100)
2	2.39	Syringaldehyde	C_9_H_9_O_4_^−^	181.0501	181.0500	−0.55	151.0392 (19), 133.0287 (53), 123.0444 (100)
3	2.61	Protocatechuic acid ^a^	C_7_H_5_O_4_^−^	153.0188	153.0189	0.65	109.0288 (100)
4	4.85	Procyanidin B1 ^a^	C_30_H_25_O_12_^−^	577.1347	577.1337	−1.73	451.1010 (8), 425.0853 (10), 407.0753 (80), 289.0702 (100), 245.0806 (16), 161.0235 (15), 125.0236 (38)
5	5.13	3-*p*-Coumaroylquinic acid	C_16_H_17_O_8_^−^	337.0924	337.0915	−2.67	191.0556 (11), 173.0449 (5), 163.0395 (100), 155.0345 (3), 119.0497 (20)
6	5.45	Procyanidin B3 ^a^	C_30_H_25_O_12_^−^	577.1347	577.1337	−1.73	451.1009 (7), 425.0853 (12), 407.0753 (90), 289.0702 (100), 245.0806 (12), 161.0235 (15), 125.0236 (42)
7	5.66	(+)-Catechin ^a^	C_15_H_13_O_6_^−^	289.0712	289.0706	−2.08	245.0808 (100), 203.0704 (86), 125.0237 (60), 109.0289 (55)
8	6.03	Procyanidin trimer	C_45_H_37_O_18_^−^	865.1980	865.1957	−2.66	695.1343 (7), 577.1334 (9), 451.0993 (20), 425.0847 (22), 407.0753 (89), 289.0703 (100), 243.0287 (39), 125.0238 (62)
9	6.47	Caffeic acid ^a^	C_9_H_7_O_4_^−^	179.0345	179.0344	−0.56	135.0443 (100)
10	6.67	Procyanidin trimer	C_45_H_37_O_18_^−^	865.1980	865.1957	−2.66	577.1334 (17), 451.1007 (20), 425.0864 (22), 407.0753 (85), 289.0703 (100), 243.0287 (38), 125.0238 (72)
11	8.18	(−)-Epicatechin ^b^	C_15_H_13_O_6_^−^	289.0712	289.0706	−2.08	245.0808 (100), 203.0704 (86), 125.0237 (70), 109.0289 (62)
12	8.46	4-*p*-Coumaroylquinic acid	C_16_H_17_O_8_^−^	337.0924	337.0915	−2.67	191.0553 (1), 173.0449 (100), 163.0394 (19), 155.0344 (5), 119.0497 (5)
13	8.80	Procyanidin trimer	C_45_H_37_O_18_^−^	865.1980	865.1957	−2.66	695.1345 (4), 577.1334 (17), 451.1021 (17), 425.0860 (18), 407.0753 (85), 289.0703 (100), 125.0238 (81)
14	9.29	Unknown	C_19_H_29_O_11_	433.1704	433.1694	−1.33	181.0857 (100), 166.0623 (95)
15	9.73	Procyanidin B2 ^b^	C_30_H_25_O_12_^−^	577.1347	577.1337	−1.73	451.1010 (7), 425.0853 (9), 407.0753 (68), 289.0702 (100), 245.0806 (16), 161.0235 (15), 125.0236 (38)
16	10.01	Taxifolin 3-*O*-glucoside	C_21_H_21_O_12_^−^	465.1033	465.1020	−2.80	447.0911 (20), 437.1066 (83), 304.0529 (2), 303.0494 (27), 285.0389 (70), 275.0547 (32), 259.0598 (42)
17	11.71	Taxifolin ^a^	C_15_H_11_O_7_^−^	303.0505	303.0502	−0.99	285.0392 (48), 241.0493 (17), 217.0496 (19), 199.0392 (12), 175.0392 (27), 125.0237 (100)
18	12.36	Unknown	C_26_H_33_O_12_^−^	537.1967	537.1958	−1.59	524.0999 (1), 327.1220 (24), 315.1220 (100)
19	13.12	Taxifolin isomer	C_15_H_11_O_7_^−^	303.0505	303.0502	−0.99	285.0393 (47), 241.0493 (16), 217.0497(23), 199.0392 (15), 175.0393 (26), 125.0237 (100)
20	16.45	Dehydroxyltaxifolin	C_15_H_11_O_6_^−^	287.0556	287.0549	−2.44	259.0599 (87), 243.0651 (17), 125.0236 (100)
21	18.71	Quercetin 3-*O*-rhamnoside	C_21_H_19_O_11_^−^	447.0928	447.0915	−2.91	302.0374 (10), 301.0336 (85), 300.0261 (100), 174.9551 (30)
22	22.21	Quercetin 7-*O*-glucoside	C_21_H_19_O_12_^−^	463.0877	463.0864	−2.81	301.0338 (100), 174.9551 (9)
23	25.08	Quercetin ^b^	C_15_H_9_O_7_^−^	301.0349	301.0340	−2.99	273.0389 (10), 178.9976 (50), 151.0028 (100)

^a^ Confirmed using available standards. ^b^ Confirmed using the in-house database.

**Table 2 antioxidants-11-02497-t002:** Concentrations and quality parameters of phenolic compounds in PBE.

Retention Time (min)	Phenolic Compound	Concentration (mg/g) ^a^	Calibration Curve	Correlation Coefficient (*R*)	LOD ^b^ (μg/mL)	LOQ ^c^ (μg/mL)
2.61	Protocatechuic acid	5.51 ± 0.00	y = 10448.33x + 1472.27	0.9997	0.03	0.08
4.85	Procyanidin B1	13.40 ± 0.18	y = 1893.37x − 61.05	0.9993	0.25	0.77
5.45	Procyanidin B3	3.75 ± 0.14	y = 1863.07x − 24.93	0.9998	0.12	0.36
5.66	(+)-Catechin	10.08 ± 0.68	y = 1995.57x + 129.22	0.9998	0.01	0.03
6.47	Caffeic acid	0.63 ± 0.01	y = 14706.33x + 205.35	0.9996	0.05	0.15
8.18	(−)-Epicatechin	0.62 ± 0.01	y = 1865.40x + 135.90	0.9997	0.09	0.27
9.73	Procyanidin B2	0.80 ± 0.00	y = 2351.27x − 89.67	0.9998	0.04	0.13
11.71	Taxifolin	12.91 ± 0.06	y = 8428.27x + 85.21	0.9999	0.04	0.13
13.12	Taxifolin isomer	2.24 ± 0.01	y = 8428.27x + 85.21	0.9999	0.04	0.13
25.08	Quercetin	0.26 ± 0.02	y = 5315.23x − 58.53	0.9998	0.09	0.26

^a^ Data are presented as mean ± standard deviation. ^b^ Limit of detection. ^c^ Limit of quantification.

## Data Availability

Not applicable.

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
