# Peer review of "Effects of Phenolic-Rich Pinus densiflora Extract on Learning, Memory, and Hippocampal Long-Term Potentiation in Scopolamine-Induced Amnesic Rats"

_antioxidants, 2022, doi:10.3390/antiox11122497_

Round 1

Reviewer 1 Report

This is an interesting study. The following issues should be addressed.

1. Please do not refer to dementia rats; scopolamine causes cognitive injury but not dementia.

2. Please do not use the word sacrifice.

3. Fig. 2E. Please show data for all 4 quadrants so that spatial bias can be assessed.

Author Response

Response: We greatly appreciated the reviewer’ valuable comments and suggestions that helped improve the quality of the manuscript.

1. Please do not refer to dementia rats; scopolamine causes cognitive injury but not dementia.

- The word dementia was changed to amnesia (or amnesic), accordingly (lines 23, 525, 580, 636, and 646)

2. Please do not use the word sacrifice.

- The word “sacrifice” was deleted.  (line 209 and Fig. 1)

3. Fig. 2E. Please show data for all 4 quadrants so that spatial bias can be assessed.

- We newly added Fig. S7 to the Supplementary file for data intuitiveness and conciseness.

Reviewer 2 Report

In the present paper, authors describe the effects of Phenolic-rich Pinus densiflora Extract on Learning  and Memory. The manuscript is original and interesting. Introduction and methods are well described, and references are relevant. However, the manuscript lacks a clear discussion; authors should separate “results” and “discussion” to better highlight their findings. Please, improve also “conclusions”.

Author Response

Reviewer #2: In the present paper, authors describe the effects of Phenolic-rich Pinus densiflora Extract on Learning and Memory. The manuscript is original and interesting. Introduction and methods are well described, and references are relevant. However, the manuscript lacks a clear discussion; authors should separate “results” and “discussion” to better highlight their findings. Please, improve also “conclusions”.

Response: Thank you for your valuable comments and suggestions. We have separated the results section and the Discussion section accordingly. In addition, the Discussion section (lines 523-530 and 634-639) and the Conclusions section (lines 641-652) have been revised or augmented to highlight our findings.

Reviewer 3 Report

It is extremely difficult for me to express an opinion on the research presented in the manuscript, mainly for ethical reasons, but also for substantive reasons. Unfortunately, I am not a supporter of research on mice or rats models in the case of Alzheimer's disease. According to:

https://www.nature.com/articles/d42473-022-00118-w

we know that:

“There are currently no animal models that fully recapitulate Alzheimer’s disease as we see it in humans,”

There is a big difference between rodents and humans, and in the case of this very complex disease, it is difficult to talk about a good animal model.

For this reason, the research presented by the authors seems pointless to me. Unnecessarily tormented animals and no result for the future. In addition, in the discussion the authors admit that:

·         However, the differences of cognitive improvement effects of behavioral tests and biochemical indices in the rat hippocampus between 15 and 30 mg/kg BW/day dosages were not completely consistent. Also, the increase in LTP induction by PBE in the hippocampal slices did not show a dose-dependent trend in this study.

(...) Thus, considering the wide range of bioavailability of phenolic compounds in PBE in ex vivo and in vivo conditions, it is hard to clearly explain why dose-dependent results were not obtained in this study. Another reason for some discrepancies in results of this study between in vivo behavioral tests and ex vivo LTP measurements may be due to differences in the age of the animals used.

Author Response

Reviewer #3: It is extremely difficult for me to express an opinion on the research presented in the manuscript, mainly for ethical reasons, but also for substantive reasons. Unfortunately, I am not a supporter of research on mice or rats models in the case of Alzheimer's disease. According to:

https://www.nature.com/articles/d42473-022-00118-w

we know that:

“There are currently no animal models that fully recapitulate Alzheimer’s disease as we see it in humans,”

There is a big difference between rodents and humans, and in the case of this very complex disease, it is difficult to talk about a good animal model.

For this reason, the research presented by the authors seems pointless to me. Unnecessarily tormented animals and no result for the future. In addition, in the discussion the authors admit that:

However, the differences of cognitive improvement effects of behavioral tests and biochemical indices in the rat hippocampus between 15 and 30 mg/kg BW/day dosages were not completely consistent. Also, the increase in LTP induction by PBE in the hippocampal slices did not show a dose-dependent trend in this study.

(...) Thus, considering the wide range of bioavailability of phenolic compounds in PBE in ex vivo and in vivo conditions, it is hard to clearly explain why dose-dependent results were not obtained in this study. Another reason for some discrepancies in results of this study between in vivo behavioral tests and ex vivo LTP measurements may be due to differences in the age of the animals used.

Response: Thank you for your valuable comments. As noted in the reviewer’s comments, we agree that no animal model perfectly matches Alzheimer’s disease (AD) in humans. Nevertheless, like our study, scopolamine is usually used in neuroscience as a model for AD-like cognitive dysfunction. Scopolamine* induces cognitive disorder by cholinergic dysfunction in rodents. As one of the most raised pathological hypotheses in the context of AD is the cholinergic theory, the application of scopolamine to AD-like model study in vivo is considered appropriate.

*Scopolamine, a Toxin-Induced Experimental Model, Used for Research in Alzheimer’s Disease (https://doi.org/10.2174/1871527319666200214104331)

There are no animal models# that structurally and quantitatively reflect complex human AD. And because animal models may or may not underlie human AD, they may not be ethical or practical in AD research. Nevertheless, the reason for using an imperfect animal model is that animal studies provide a better pharmacological understanding of the disease by selecting potential targets for AD through animal testing prior to clinical trial. Our study elucidated some mechanisms of AD with limited animal models, which might be the basis for understanding AD or establishing new hypotheses. As the reviewer noted, we are aware of the limitations of our study and will investigate further to make it better.

#Animal models of Alzheimer’s disease and drug development (https://doi.org/10.1016/j.ddtec.2012.04.001)

Reviewer 4 Report

This is an interesting and innovative manusvript that should be published

Author Response

Reviewer #4: This is an interesting and innovative manuscript that should be published

Response: We are greatly appreciate for reviewing our paper.